# Combined Use of the Ab105-2φΔCI Lytic Mutant Phage and Different Antibiotics in Clinical Isolates of Multi-Resistant *Acinetobacter baumannii*

**DOI:** 10.3390/microorganisms7110556

**Published:** 2019-11-12

**Authors:** Lucia Blasco, Anton Ambroa, Maria Lopez, Laura Fernandez-Garcia, Ines Bleriot, Rocio Trastoy, Jose Ramos-Vivas, Tom Coenye, Felipe Fernandez-Cuenca, Jordi Vila, Luis Martinez-Martinez, Jesus Rodriguez-Baño, Alvaro Pascual, Jose Miguel Cisneros, Jeronimo Pachon, German Bou, Maria Tomas

**Affiliations:** 1Microbiology Department-Research Institute Biomedical A Coruña (INIBIC), Hospital A Coruña (CHUAC), University of A Coruña (UDC), 15495 A Coruña, Spain; lucia.blasco@gmail.com (L.B.); anton17@mundo-r.com (A.A.); maria.lopez.diaz@sergas.es (M.L.); laugemis@gmail.com (L.F.-G.); bleriot.ines@gmail.com (I.B.); trastoy.rocio@gmail.com (R.T.); german.bou.arevalo@sergas.es (G.B.); 2Microbiology Department-Research Institute Biomedical Valdecilla (IDIVAL), Hospital Marques de Valdecilla, 39008 Santander, Spain; jvivas@idival.org; 3Laboratory of Pharmaceutical Microbiology, Ghent University, 9000 Gent, Belgium; Tom.Coenye@UGent.be; 4Clinical Unit for Infectious Diseases, Microbiology and Preventive Medicine, Hospital Universitario Virgen Macarena/Department of Microbiology and Medicine, University of Seville/Biomedicine Institute of Seville (IBIS), 41009 Seville, Spain; felipefc@us.es (F.F.-C.); jesusrb@us.es (J.R.-B.); apascual@us.es (A.P.); 5Institute of Global Health of Barcelona (ISGlobal), Hospital Clínic-Universitat de Barcelona, 170, 08036 Barcelona, Spain; jvila@ub.edu; 6Unit of Microbiology, University Hospital Reina Sofía, Department of Microbiology, University of Córdoba, Maimonides Biomedical Research Institute of Cordoba (IMIBIC), 14004 Cordoba, Spain; luis.martinez.martinez.sspa@juntadeandalucia.es; 7Clinical Unit for Infectious Diseases, Microbiology and Preventive Medicine, Hospital Universitario Virgen del Rocío/Department of Microbiology and Medicine, University of Seville/Biomedicine Institute of Seville (IBIS), 41009 Seville, Spain; jmcisnerosh@gmail.com (J.M.C.); pachon@us.es (J.P.)

**Keywords:** *Acinetobacter baumannii*, multiresistant, mutant lytic phage, phage therapy, antibiotic-phage synergy

## Abstract

Phage therapy is an abandoned antimicrobial therapy that has been resumed in recent years. In this study, we mutated a lysogenic phage from *Acinetobacter baumannii* into a lytic phage (Ab105-2phiΔCI) that displayed antimicrobial activity against *A. baumannii* clinical strain Ab177_GEIH-2000 (isolated in the GEIH-REIPI Spanish Multicenter *A. baumannii* Study II 2000/2010, Umbrella Genbank Bioproject PRJNA422585, and for which meropenem and imipenem MICs of respectively, 32 µg/mL, and 16 µg/mL were obtained). We observed an in vitro synergistic antimicrobial effect (reduction of 4 log–7 log CFU/mL) between meropenem and the lytic phage in all combinations analyzed (Ab105-2phiΔCI mutant at 0.1, 1 and 10 MOI and meropenem at 1/4 and 1/8 MIC). Moreover, bacterial growth was reduced by 8 log CFU/mL for the combination of imipenem at 1/4 MIC plus lytic phage (Ab105-2phiΔCI mutant) and by 4 log CFU/mL for the combination of imipenem at 1/8 MIC plus lytic phage (Ab105-2phiΔCI mutant) at both MOI 1 and 10. These results were confirmed in an in vivo model (*G. mellonella*), and the combination of imipenem and mutant Ab105-2phiΔCI was most effective (*p* < 0.05). This approach could help to reduce the emergence of phage resistant bacteria and restore sensitivity to antibiotics used to combat multi-resistant strains of *Acinetobacter baumannii.*

## 1. Introduction

Multi-drug resistant (MDR) bacteria, such as *A. baumannii* are considered to be a major concern by the World Health Organization (WHO), because of their ability to acquire antimicrobial resistance via intrinsic characteristics and mechanisms (e.g., presence of the outer membrane) or via mechanisms acquired by horizontal genetic transfer [1,2]. This situation has led to an urgent need to develop new antimicrobial agents and to a renewed interest in phage therapy. Phage therapy was first developed in the 1920s but was abandoned in the Western world after the discovery of antibiotics. However, the use of phage therapy continued in Eastern countries, such as Poland and USSR, where bacteriophages are used for the prophylaxis and treatment of infections, such as dysentery, ulcers, and methicillin resistant *Staphylococcus aureus* (MRSA) infections [3,4].

Phage therapy is now considered a real option for treating MDR bacteria, and there are some examples of its use in treating human patients [5]. Phages are bacterial viruses, and like other viruses, they are obligate parasites that enter host cells through mechanisms that are based on receptor recognition. Genetic material is then injected into the bacteria and use the bacterial machinery to produce phage proteins [6,7]. Phages generally undergo a lytic (virulent) or lysogenic (temperate) life cycle. Lytic phages infect, and rapidly lyse and kill host cells, releasing phage progeny into the surrounding medium. Lysogenic phages infect the host cell and integrate their nucleic acid into the host genome, or exist as plasmids in the host cells, remaining in a stable prophage state for generations. Prophages can be “induced” to exit the cell as lytic phages under some conditions, such as the presence of antibiotics [8,9]. The lysogenic/lytic cycle of temperate bacteriophages is controlled by Cro, CI, and CII proteins; the Cro protein induces the lytic state and the CI repressor protein inhibits the Cro protein, thereby inducing the lysogenic state [10].

Only lytic phages are used in phage therapy as lysogenic phages can transfer resistance genes or virulence factors to the host [11].

The combined use of antibiotics and phages has been tested in several studies, demonstrating strong control of the bacteria, and a reduction in the development of phage and/or antibiotic resistance [12,13]. Phages are good candidates for use in combination with antibiotics for various reasons, including that they have a different mechanism of action from antibiotics; hold a narrow spectrum of activity, which protects the normal microbiota; they can multiply at the infection site; they are abundant in nature and can be easily isolated; and production costs are low [14,15,16].

In this study, we produced a mutant lytic phage from a lysogenic phage, that is incorporated in the genome of a clinical strain of *A. baumannii* by deleting the CI repressor gene, and thus, preventing the entry of the phage into the lysogenic cycle [10,17]. We then tested the antimicrobial activity of the novel lytic phage, Ab105-2phiΔCI, in combination with carbapenem antibiotics (meropenem and imipenem) against a carbapenem-resistant strain of *A. baumannii*. The combined therapy enhanced the antimicrobial activity of both, the phage and the antibiotic; the bacterium became sensitive to the antibiotics and the emergence rate of phage resistant bacteria was reduced.

## 2. Material and Methods

### 2.1. Bacterial Strains

In this study, we used 20 clinical strains isolated from Spanish hospitals during the GEIH-REIPI Spanish Multicenter *Acinetobacter baumannii* Study II 2000–2010, GenBank Umbrella project PRJNA422585 (https://www.ncbi.nlm.nih.gov/bioproject) (Table 1).

### 2.2. Obtaining the Lytic Phage Mutant

The bacteriophage sequence Ab105-2phi (Genbank: KT5880759) detected in clinical strain *A. baumannii* Ab105GEIH_2010 was analyzed and the CI gene identified as ORF 17. The CI gene was deleted by double homologous recombination with the suicide vector pMo130TelR [18,19]. The primers were first designed for the amplification of the flanking regions (1000 bp) of the CI gene. These regions were amplified by PCR and ligated and cloned into the pMo130telR vector (Table 2). This construction was transformed in *Escherichia coli* DH5α to produce large numbers of the plasmid with the gene flanking regions. The plasmid was purified and transformed in the *A. baumannii* Ab105 clinical strain by electroporation, and incubated for two hours at 37 °C without antibiotic, thereby producing a recombinant wild type with the mutated gene integrated in its genome. Finally, the mutants were selected in the presence of kanamycin (50 µg/mL). In order to isolate only those mutants with the CI gene deletion in the chromosome, the plasmid loss was induced in the absence of kanamycin, and the recombinant clones were selected in the presence of 15% sucrose. In order to isolate the mutant phage Ab105-2phiΔCI from the bacterial clones by which the CI gene was deleted, a selected clone was incubated in LB broth, which is supplemented with mitomycin (10 ug/mL) to induce release of the phages. The supernatant was collected, treated with chloroform, and filtered (20 µm). The filtered supernatant was used to infect the clone without the phage, and plaques were obtained by the agar overlay method [20]. A clear plaque was isolated by PCR and sequencing was conducted to confirm the correct deletion of the CI gene.

A clone of strain Ab105GEIH_2010, induced with mitomycin, was isolated and excision of the phage was confirmed by PCR of the CI gene and the flanking regions (1000 pb each region) of the gene.

### 2.3. Host Range and Efficiency of Plating Analysis

The host range of the lytic mutant phage Ab105-2phiΔCI was established by applying the spot test [21] to the 20 clinical strains of *A. baumannii* under study. Efficiency of Plating (EOP) was established as the ratio between the test strain titre and the host strain titre [22].

### 2.4. Transmission Electron Mmicroscopy (TEM) and Live-Cell Imaging

A broth culture of strain Ab177_GEIH-2000 was infected with the lytic mutant phage Ab105-2phiΔCI. The lysates were centrifuged at 3400× *g* for 10 min and the supernatant was filtered through a 0.22 µm filter (Merck Millipore, Ltd. Tullagreen, Carrigtwahill, Co Cork, Ireland). NaCl was added to a final concentration of 0.5 M, and the suspensions were mixed thoroughly and left on ice for 1 h. The suspensions were centrifuged at 3400× *g* for 40 min at 4 °C, and the supernatants were transferred to sterile tubes. PEG 6000 (10% *w/v*) was added, dissolved, and incubated overnight at 4 °C. Bacteriophages were then precipitated at 3400× *g* for 40 min at 4 °C and resuspended in SM buffer (0.1 M NaCl, 1 mM MgSO4, 0.2 M Tris-HCl, pH 7.5) [23]. The samples were negatively stained with 1% aqueous uranyl acetate before examination by electron microscopy.

Live-cell imaging was carried out by time-lapse microscopy after initial adsorption of the mutant lytic phage Ab105-2phiΔCI to the clinical strain Ab177_GEIH-2000 at 37 °C in agar slices, which were placed directly between stainless steel O-rings. The use of extracellular DNA markers enabled the lysis of more than 300 bacteria to be monitored in real time.

### 2.5. Adsorption Curve, One Step Growth Curve, and Infection Curve

An overnight culture of *A. baumannii* clinical strain Ab177_GEIH-2000 was diluted 1:100 in LB broth, and incubated at 37 °C at 180 rpm, until an early logarithmic phase, i.e., at an optical density of 0.2 (OD 600nm). At this point the culture was infected with the lytic mutant phage Ab105-2phiΔCI at a multiplicity of infection (MOI) of 0.1. The adsorption curve and the one step growth curve were determined after growing the phage in LB, supplemented with CaCl_2_, as previously described [20,24]. In the one step growth curve, the latent period was defined as the interval between adsorption of the phages to the bacterial cells and the release of phage progeny. The burst size of the phage was determined as the ratio of the final number of free phage particles to the number of infected bacterial cells during the latent period [22].

An early exponential culture of the strain Ab177_GEIH-2000 in LB, supplemented with CaCl_2_, was infected with the lysogenic phage Ab105phi2 and the mutant lytic phage Ab105phi2ΔCI at different MOIs (0.1, 1 and 10), and the corresponding infection curves were constructed. The phage cultures were maintained at room temperature during the adsorption period and then incubated at 37 °C and 180 rpm for 6 h. The optical density was measured at intervals of one hour during this period.

### 2.6. Frequency of Occurrence of Phage Resistant Bacteria

Phage resistant mutants were produced as previously described [25]. To determine the emergence of phage resistant mutants, an overnight culture of strain Ab177_GEIH-2000 was diluted 1:100 in LB and grown to an OD600nm of 0.6–0.7. An aliquot of 100 µL of the culture containing 10^8^ colony forming units (CFU)/mL was serially diluted, and each dilution mixed with 100 µL of 10^9^ plaque forming units (PFU)/mL, and plated by the agar overlay method [20]. The plates were incubated at 37 °C for 18h and the number of CFUs was counted. The same procedure was used to produce phage resistant mutants in the presence of the antibiotics doxycycline, meropenem, or imipenem, which were added to the plates, each at 25% of the minimum inhibitory concentration (MIC). The frequency of occurrence of phage resistant mutants and phage-antibiotic resistant mutants was calculated by dividing the number of resistant bacteria by the total number of sensitive bacteria.

### 2.7. Antimicrobial Activity of the Mutant Lytic Phage Ab105-2phiΔCI in Biofilm

An overnight culture of the *A. baumannii* clinical strain Ab177_GEIH-2000 was diluted 1:100 and used to inoculate 100 µL of LB in some wells of a 96 multi-well plate. The plate was maintained at 37 °C in static conditions for 4 h. The medium was then discarded and the wells were washed twice with PBS before 100 µL of fresh LB was added. After 24 h at 37 °C, the medium was again discarded and the wells were washed with PBS, and filled with 90 µL of SM buffer, then 10 µL of phage Ab105-2phiΔCI (10^7^ PFU/mL) was added. SM buffer (100 µL) was added to control wells. The plates were then incubated at 37 °C for 24 h. Finally, the supernatant was discarded and the wells were washed with PBS. Half of the wells were used to quantify the CFUs and the other half were used to quantify the biofilm. PBS (100 µL) was added to the wells used to quantify the CFUs and the plates were agitated for 5 min and sonicated for another 5 min. The suspension was serially diluted and plated on LB plates. For quantification of the biofilm, 100 µL of methanol was added to each well and discarded after 10 min. Once the methanol had completely evaporated, 100 µL of crystal violet (0.1%) was added and discarded after 15 min. Finally, the wells were washed with PBS before the addition of 150 µL of acetic acid (30%), and the absorbance was measured at OD 595 nm.

### 2.8. Antimicrobial Activity in Combination with Antibiotics

A bacterial killing assay was constructed to determine the synergy of phage Ab105-2phiΔCI in combination with meropenem, imipenem and doxycycline at 1/8 and 1/4 of the respective MICs (meropenem 32 µg/mL, imipenem 16 µg/mL and doxycycline 64 µg/mL). An overnight culture of the tested strain was diluted at 1:100 in LB broth supplemented with 10uM CaCl_2_ and incubated at 37 °C and 180 rpm until the culture reached an early exponential phase at 0.2 OD (600nm). At this point, antibiotic and the Ab105-2phiΔCI phage were added to the culture. The flasks were maintained at room temperature during the adsorption period before being incubated at 37 °C and 180 rpm for 24 h. Aliquots were removed after 6 h and 24 h and were serially diluted and plated in LB plates for subsequent counting of CFU.

### 2.9. Galleria mellonella Survival Assay

The *Galleria mellonella* model used was an adapted version of a previously developed model also used to study bacteriophage therapy [26,27]. The procedure was as follows: twelve *G. mellonella* larvae, acquired from TruLarvTM (Biosystems Technology, Exeter, Devon, UK), were each injected in the left proleg with 10 μL of a suspension of *A. baumannii* Ab177_GEIH-2000, diluted in sterile phosphate buffer saline (PBS) containing 1 × 10^5^ CFU (± 0. 5 log). The injection was performed with a Hamilton syringe (volume 100 μL) (Hamilton, Shanghai, China). One hour after infection, the larvae were injected in the right proleg with 10 µL of the lytic mutant phage Ab105-2phiΔCI, at MOI 10, in combination with meropenem at 1/4 MIC and imipenem at ¼ MIC. The controls included 10 µL of the lytic mutant phage Ab105-2phiΔCI at MOI 10, or meropenem at 1/4 MIC and imipenem at 1/4 MIC. The injected larvae were placed in Petri dishes and incubated in darkness at 37 °C. The number of dead larvae was recorded after 72 h. The larvae were considered dead when they showed no movement in response to touch [26].

The survival curves for the in vivo *G. mellonella* infection model were constructed using GraphPad Prism v.6 (San Diego, CA, USA), where the data were analyzed using the Log-Rank (Mantel-Cox, City, State if USA, Country) test. The data were expressed as mean values, and the differences were considered statistically significant at *p* < 0.05.

## 3. Results

### 3.1. Obtaining the Lytic Mutant of the Phage Ab105phi-2ΔCI

After deleting the CI gene from the temperate phage Ab105-phi2, as previously reported in *Salmonella* [17], we obtained a lytic mutant, designated Ab105-phi2ΔCI, which produced characteristic clear lytic plaques. This is in contrast with the turbid plaques produced by the temperate Ab105-phi2 phage (Figure 1A1). PCR of the DNA, isolated from the Ab105-2phiΔCI phage, confirmed the deletion of the CI gene. PCRs were conducted with the CI genes and combinations of these primers with those of the flanking regions, confirming that no amplification was obtained. PCRs with the primers (UPCI[NotI]Fw/DWCI[SphI]Rev) of the flanking regions of the gene CI were also conducted, and the expected region of 2000 pb was obtained (size without the CI gene). Finally, this amplicon was sequenced and the CI gene deletion was confirmed. Excision of the phage was also confirmed in a clone induced with mitomycin, as no positive PCR were obtained with the CI primers or with the flanking region primers.

Infection curves for the temperate phage Ab105-2phi and the lytic mutant phage Ab105-2phiΔCI were constructed and compared, showing that the lytic mutant killed the culture at all MOI levels tested, as reflected by a large decrease in the optical density. Although, a reduction in growth was observed when the culture was infected with the lysogenic phage Ab105-2phi, the decrease was less than with the lytic mutant. In both cases, the reduction in growth was first observed at MOI 10, but regrowth was also first observed at this MOI, probably due to the emergence of resistance (Figure 1B).

### 3.2. Morphology and Host Range of the Lytic Mutant Phage Ab105-phi2ΔCI

The lytic mutant Ab105-2phiΔCI was isolated and the virion morphology was observed by TEM, revealing that this phage has the typical structure of the Siphoviridae as the wild type phage Ab105-phi2 [28]. All plaques were transparent and about 1mm in diameter (Figure 1A2).

The lytic spectrum of activity of the mutant phage Ab105-2phiΔCI covered 25% of the clinical strains of *A. baumannii* tested. The strain Ab177_GEIH-2000 yielded the highest EOP (1.55) (Table 1). This strain was thus selected for further assays.

### 3.3. Adsorption and One Step Growth Curve

Both, the adsorption and the one step growth curve were established using host strain Ab177_GEIH-2000, as the EOP of this strain was the most appropriate and also because this strain does not have complete prophages, as previously determined [28]. The adsorption time (12 min) was determined in order to establish the one step growth curve, which revealed a latent period of 30 min and a burst size of approximately 32 ± 2 PFU per infected cell (Figure 1C).

### 3.4. Antimicrobial Activity of the Mutant Lytic Phage Ab105-2phiΔCI on Biofilm

Biofilm was produced with the clinical strain of *A. baumannii* Ab177_GEIH-2000 susceptible to the mutant lytic phage Ab105-phi2ΔCI. The treatment of the biofilm with 10^7^ PFU of this lytic mutant phage caused a statistically significant reduction in the biofilm biomass. The antimicrobial activity against the biofilm forming bacteria was confirmed by a decrease in the CFU, quantified in the presence of the mutant lytic phage (Figure 1D).

Finally, the lytic activity of the mutant phage can be observed in Video 1 (Appendix A).

### 3.5. Determination of the Emergence Rate of Phage Resistant Mutants

Strain Ab177_GEIH-2000 was resistant to meropenem, imipenem and doxycycline (MICs: Meropenem 32 µg/mL, imipenem 16 µg/mL, and doxycycline 64 µg/mL). In all cases the combination of the phage and antibiotic reduced the rate of emergence of phage-resistant mutants, relative to the rate of resistant mutants in the presence of the phage alone (Table 3).

### 3.6. Effect of the Combination of Phage and Antibiotic on the Bacterial Killing Assays

Bacterial killing assays were constructed for *A. baumannii* clinical strain Ab177_GEIH-2000 in the presence of a combination of the lytic mutant phage Ab105-2phiΔCI at different MOIs (0.1, 1, and 10) and three antibiotics (at 1/4 and 1/8 MIC) to which Ab177_GEIH-2000 is resistant: Meropenem, imipenem, and doxycycline (Figure 2).

A reduction in the number of CFU was observed with the phage at both MOI 1 (4 log) and MOI 10 (1 log) after 6 h, but no differences from the control were observed after 24 h. The reduction was even greater when the phage was combined with meropenem or imipenem (both carbapenems).

For meropenem plus phage, a synergistic effect was observed after 6 h for all combinations (from 4 log to 7 log CFU/mL). The growth of the *A. baumanni* strain was similar to control levels after 24 h for all concentrations of meropenem plus phage at MOI 1. The synergistic effect was only maintained with the combination of meropenem at 1/4 MIC and phage Ab105-2phiΔCI at MOI10, yielding a difference in bacterial growth of 6 log CFU/mL, relative to that corresponding to the meropenem control (Figure 2A1,2A2).

As with meropenem, the combination of different concentrations of imipenem and the lytic mutant phage had a synergistic effect after 6 h in all cases, with a reduction in bacterial growth of 8 log CFU/mL for the combination of imipenem at 1/4 MIC, plus phage, and 4 log CFU/mL for the combination of imipenem at 1/8 MIC plus phage. The synergistic effect was maintained for 24 h in the combinations of imipenen at 1/4 MIC, with phage at MOI1 and MOI10, but not in the combinations of imipenem at 1/8 MIC and both phage concentrations (Figure 2B1,2B2).

No synergistic effects were observed with doxycycline, and the combination had no more effect than the phage alone at MOI 1. However, when the combinations included the phage at MOI 10, a slight decrease in the CFU count was observed (less than 1 log CFU/mL), independently of the antibiotic concentration (Figure 2C1,2C2).

The curves obtained for the lytic mutant phage controls showed that Ab177_GEIH-2000 grew at control rates after 24h, due to the acquisition of phage resistance. However, the growth was higher at MOI 10, than at MOI 1 after 6 h, probably because resistance emerges faster at this MOI than at lower MOI.

### 3.7. Galleria mellonella Survival Assays in the Presence of Meropenem and Imipenem in Combination with the Lytic Mutant Phage Ab105-phi2ΔCI

The combinations of antibiotic and the phage Ab105-2phiΔCI, that resulted in the reduction of the CFU of Ab177_GEIH-2000 at 24h in vitro were assayed in a *G. mellonella* (wax moth) larvae survival model (Figure 3). When the infected larvae were treated with imipenem and mutant lytic phage Ab105-2phiΔCI, the survival rate was found to be statistically significantly higher than the larvae treated with the antibiotic or the phage alone and of untreated larvae (*p* < 0.05). Similar results were obtained for meropenem but in this case. Although, larval survival was higher after the combinatory treatment than after phage only or no treatment, the difference relative to meropenem alone was not statistically significant (*p* = 0.2183). This was probably due to the higher MIC of meropenem than of imipenem (32 µg/mL *versus* 16 µg/mL) for the Ab177_GEIH-2000 strain, indicating the need to administer greater amounts of mutant lytic phage Ab105-2phiΔCI.

## 4. Discussion

Lytic phages are widely used in phage therapy, but temperate or lysogenic phages have not generally been considered suitable for the purpose, because they can enhance host competence and survival. However, temperate phages are present in almost half of bacteria that have been sequenced. Phages that are specific to pathogens causing infections can be easily identified. In addition, the problems caused by horizontal genetic transfer can now be avoided due to next generation sequencing, which enables phages to be selected, that do not pose a risk of transferring undesirable genes, such as endotoxins [15]. Temperate phages can also be easily engineered in their lysogenic state for use in phage therapy, by different means: By modifying the genes of interest as phage receptors to extend the host range; by inhibiting the lytic ability of phages without the release of endotoxins; modifying genes to enhance the killing effect of bacteriophages; increasing the life time of phages in the circulatory system of mammalians, and; transforming lysogenic phages into lytic phages [17,29,30,31,32,33].

In this study, we selected a temperate phage, Ab105-2phi, which did not have any toxic or virulence genes in its genome (Figure 1A1). This phage was selected with the objective of converting it into a lytic phage with potential use in phage therapy. The technique was previously described in *Salmonella enterica* bacteriophage SPN9CC and in the mycobacteriophage BPs33ΔHTH_HRM10, recently used in a phage cocktail to treat a patient with a disseminated drug-resistant *Mycobacterium abscessus* [17,34]. The technique is based on the deletion of the CI repressor gene, which encodes the CI protein and binds to two operators, thereby repressing the *Cro* gene required for lytic development. Deleting the CI gene thus maintains the phage in a lysogenic state [10,17].

Conversion of the lysogenic into a lytic phage was confirmed, first by PCR and sequencing. Then, by the presence of clear plaques and by the infection curves for both phages: Lysogenic Ab105-phi2 and the lytic mutant Ab105-phi2ΔCI. The differences in optical density in both cases confirmed the production of a lytic mutant, and the emergence of phage resistant mutants for both phages. At MOI 10, the inhibition of growth was greater and occurred earlier than at other MOI, but resistant bacteria emerged earlier than at lower MOI, as also observed by other authors [35]. In addition, this effect was observed in the bacterial killing assays, where the growth at 6 h was greater at MOI10 than at MOI1. The mutant lytic phage also presented a latent period of 30 min, and a moderate burst size of 32 ± 2 PFU per cell was obtained with the mutant lytic phage, values in the range of those obtained in several studies for different lytic phages, including phages from *A. baumannii* [5,36,37,38,39]. The burst size is inversely related to the risk of emergence of phage resistant bacteria [40], which is one of the main objectives of phage therapy research, commonly addressed by the use of phage cocktails [41].

Although the antimicrobial activity of this mutant lytic phage was established by its ability to reduce the absorbance in a bacterial culture and also to reduce the biofilm biomass, any reduction in the development of phage resistance would increase its potential use as a therapeutic phage. In this case, the strategy we combined the phage with antibiotics [13] to enhance the potential of the Ab105-2phiΔCI phage as a therapeutic phage, and observed an almost 1 log reduction in the emergence of phage resistant mutants in the presence of the antibiotics assayed. The synergistic effect (almost 2 log decrease between the combined therapy and the compounds alone) resulted from the combination of the lytic mutant phage Ab105-2phiΔCI, and meropenem or imipenem enhanced the bactericidal effect of both the antibiotic and the phage. A strong antimicrobial effect was obtained by combining the phage at a high MOI and the antibiotic at concentrations much lower than the MIC, thereby restoring the sensitivity of the strain to imipenem and meropenem. As the host strain does not possess beta-lactamases, the resistance is probably due to the action of a Resistance-Nodulation-Division (RND) efflux pump, containing proteins that can act as phage receptor proteins. Therefore, the phage blocks the efflux pump and the antibiotic sensitivity of the strain would thus be increased [42]. The activity of the efflux pump explains the differences between antibiotics, as the efflux pumps that expulse carbapenems can act on doxycycline, when present at low levels [43].

The increase in the antimicrobial activity when the carbapenem antibiotics and the mutant lytic phage were used together was also confirmed in the survival assays with *G. mellonella*, as survival was higher in larvae that received the combined treatments. However, when the combination included meropenem (MIC, 32 µg/mL), survival was not statistically significantly higher, indicating that administration of a larger number of mutant lytic phage Ab105-2phiΔCI would be necessary (in vivo).

In conclusion, this is the first in vitro and in vivo study by which a mutant lytic phage has been used in combination with carbapenem antibiotics (imipenem and meropenem). This reduces the emergence of resistance to the phages and restores the sensitivity to antibiotics, thereby increasing the therapeutic potential of the phage. The conversion of temperate phages (with a known genomic profile) into lytic phages may provide a new source of phages for use in phage therapy.

## Figures and Tables

**Figure 1 microorganisms-07-00556-f001:**
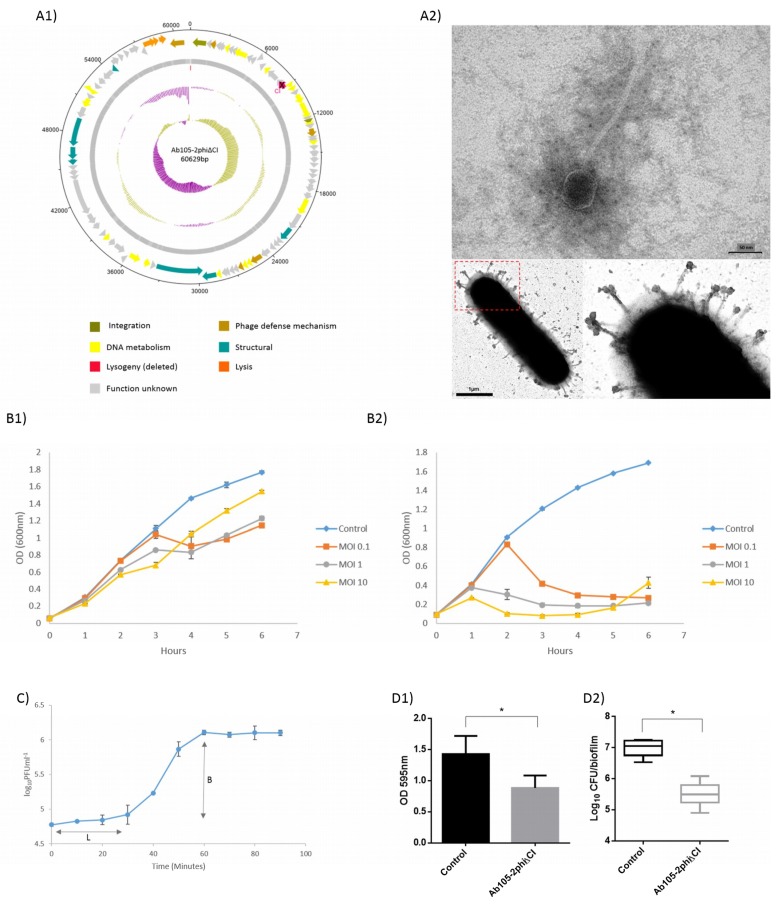
Graphical representation of the Ab105-2phiΔCI phage. The ORF and direction of transcription are indicated by arrows. (**A1**) The protein functions are indicated in different colours, and the GC content and GC skew are shown as pink and green circles respectively. (**A2**) TEM image of the mutant lytic phage Ab105-phi2ΔCI and mutant lytic phage Ab105-phi2ΔCI attached to the cell surface. (**B1**) Infection curves for the lysogenic phage Ab105-2phi and (**B2**) the mutant lytic phage Ab105-2phiΔCI. (**C**) One step growth curve of the mutant lytic phage Ab105-phi2ΔCI (L: Latent period; B: burst size). Mutant lytic phage Ab105-phi2ΔCI antibiofilm activity on the biofilm produced by the clinical strain of *A. baumannii* Ab177_GEIH-2000. (**D1**) Reduction in the biofilm and reduction in the number of CFUs present in the biofilm after treatment with the mutant lytic phage Ab105-phi2ΔCI (**D2**). Figures B, C and D show the mean values +/− SD from three independent assays. Statistically significant differences (*p* < 0.05) were determined by t-Student test (GraphPad Prism v.6).

**Figure 2 microorganisms-07-00556-f002:**
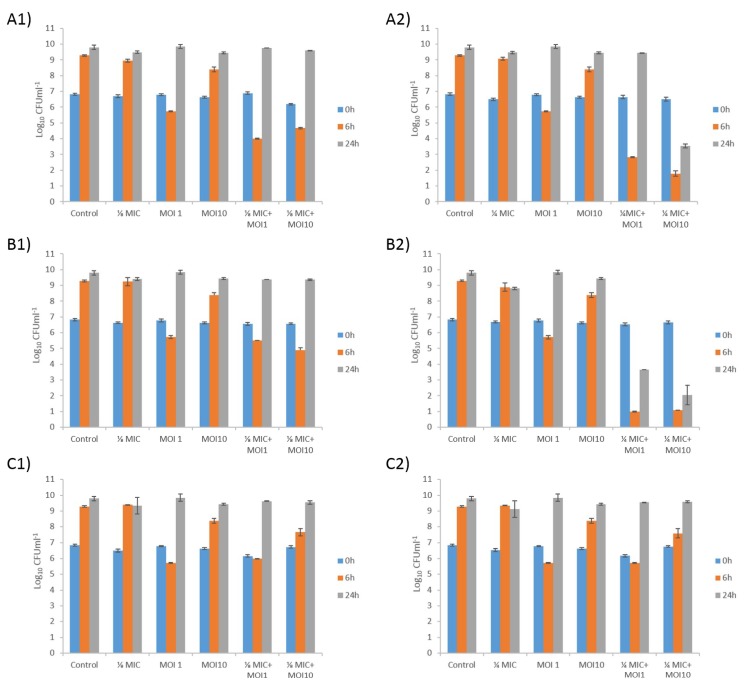
Bacterial killing assays for *A. baumannii* clinical strain Ab177_GEIH-2000 determined using the mutant lytic phage Ab105-2phi∆CI at MOI 1 and MOI10 in combination with meropenem at (**A1**) 1/8 MIC and (**A2**) 1/4 MIC;(**B1**) imipenem at 1/8 MIC and (**B2**)1/4 MIC, and (**C1**) doxycycline at 1/8 MIC and (**C2**)1/4 MIC. Values shown in the graphs are the means +/− SD from three independent assays.

**Figure 3 microorganisms-07-00556-f003:**
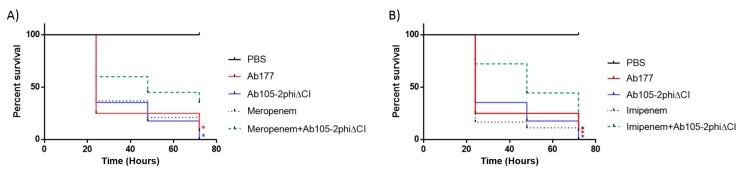
*G. mellonella* survival 96 h after an infection with Ab177_GEIH-2000 (1 × 10^5^ CFU) treatment with mutant lytic phage Ab105-2phi∆CI (1 × 10^6^ PFU) and the antibiotics meropenem at (**A**) 1/4 MIC and imipenem at (**B**) 1/4 MIC. The Log-Rank (Mantel-Cox) test, */* (*p* < 0.05) was used to compare the combination of imipenem and meropenem plus phage (line green) with each antibiotic alone (*) or the phage alone (*); *(*p* < 0.05) for comparison of the combination of the phage (line green) and antibiotics (imipenem or meropenem) and untreated infection (*).

**Table 1 microorganisms-07-00556-t001:** Bacterial strains used in this study. Phage host range determined by spot test and efficiency of plating (EOP).

Strain	ST	Spot	EOP	Spanish Hospital Where the Strain Was Isolated
Ab105_GEIH-2010	2	+/−	1	Hospital Universitario Virgen del Rocío (Seville, Spain)
Ab192_GEIH-2000	2	+/−	0.22	Hospital Universitario Virgen del Rocío (Seville, Spain)
Ab404_GEIH-2010	80	+	0.0002	Hospital Dr. Molines (Valencia, Spain)
Ab166_GEIH-2000	2	+/−	-	Hospital Universitario Virgen del Rocío (Seville, Spain)
Ab177_GEIH-2000	2	+	1.55	Hospital Universitario Virgen del Rocío (Seville, Spain)
Ab13_GEIH-2010	79	-	-	Hospital Santiago de Compostela (Santiago de Compostela, Spain)
Ab09_GEIH-2010	297	-	-	Hospital Santiago de Compostela (Santiago de Compostela, Spain)
Ab160_GEIH-2000	2	-	-	Hospital Universitario Virgen del Rocío (Seville, Spain)
Ab155_GEIH-2000	2	-	-	Hospital Universitario Virgen del Rocío (Seville, Spain)
Ab05_GEIH-2010	186	-	-	Hospital A Coruña (A Coruña, Spain)
Ab22_GEIH-2010	52	-	-	Hospital Pontevedra (Pontevedra, Spain)
Ab421_GEIH-2010	2	-	-	Hospital Insular (Gran Canaria, Spain)
Ab77_GEIH-2000	2	-	-	Hospital Universitario Ramon y Cajal (Madrid, Spain)
Ab141_GEIH-2000	179	-	-	Complejo Hospitalario Toledo (Toledo, Spain)
Ab217_GEIH-2010	2	-	-	Hospital Reina Sofía (Cordoba, Spain)
Ab235_GEIH-2010	2	-	-	Hospital Marqués de Valdecilla (Santander, Spain
Ab37_GEIH-2010	2	-	-	Hospital Virgen del Rocío (Seville, Spain)
Ab222_GEIH-2000	181	-	-	Hospital Bellvitge (Barcelona)
Ab461_GEIH-2010	2	-	-	Hospital del Mar (Barcelona, Spain)
Ab173_GEIH-2010	88	-	-	Hospital San Agustín (Avilés, Spain)

ST: Sequence Type. Spot test: (+) clear spot; (+/−) turbid spot; (-) no spot.

**Table 2 microorganisms-07-00556-t002:** Primers used to delete the CI gene.

Primer	Sequence	Strain/Plasmid
UPCI [NotI]Fw	GGG*GCGGCCGC*TGAAGAATTCATCACTTG	Ab105_GEIH-2010
UPCI[BamHI]Rev	GGG*GGATCC*CGTTACTTCTATCGGAAT	Ab105_GEIH-2010
DWCI[BamHI]Fw	GGG*GGATCC*ATTAAGGTTTTAGGTGAT	Ab105_GEIH-2010
DWCI[SphI]Rev	GGG*GCATGC*TAAATCATCCAAATCGAC	Ab105_GEIH-2010
CIFw	ATGGACAAATTTATGGCTAC	Ab105_GEIH-2010
CIRev	TAACTTTTTCTAACACGCT	Ab105_GEIH-2010
IntCIFw	AAAGCGCTGCCAACTTTT	Ab105_GEIH-2010
IntCIRev	CAACAGATTCATCCTCAT	Ab105_GEIH-2010
pMo130TelRFw	ATTCATGACCGTGCTGAC	pMo130TelR
pMo130TelRRev	CTTGTCTGTAAGCGGATG	pMo130TelR
Plasmid	Description	Origin
pMo130TelR	Suicide plasmid, *xylE*^+^, *sac*B^+^, km^R^, Tel^R^	[19]

Restriction enzyme sites are shown in italics.

**Table 3 microorganisms-07-00556-t003:** Frequency of phage resistant mutants. Phage resistant mutant frequency in the presence of the combination of doxycycline, meropenem and imipenem at ¼ MIC in combination with lytic mutant phage Ab105-2phiΔCI (MOI 10) was calculated.

Sample	Frequency of Phage Resistant Mutants
Ab105-2phiΔCI	1.70 × 10^−6^
Ab105-2phiΔCI + Doxycycline	1.31 × 10^−7^
Ab105-2phiΔCI + Meropenem	2.10 × 10^−7^
Ab105-2phiΔCI + Imipenem	1.90 × 10^−7^

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
