# Peer review of "Combined Use of the Ab105-2φΔCI Lytic Mutant Phage and Different Antibiotics in Clinical Isolates of Multi-Resistant Acinetobacter baumannii"

_microorganisms, 2019, doi:10.3390/microorganisms7110556_

Round 1
Reviewer 1 Report
The manuscript by Blasco et al. describes the conversion of a prophage integrated in a multi-resistant Acinectobacter baumannii strain into a lytic phage. This could be achieved by deletion of its CI repressor gene. The resulting lytic phage was observed to have an antimicrobial activity again clinical A. baumannii strains in vitro and to a lesser extend in a Galleria mellonella model of infection. The concept explored in this article which include the use of phages for the treatment of A. baumannii and phage antibiotics synergism are not novel. However, only few examples exist for the lysogenic to lytic conversion of prophage and their potential use as antimicrobial agents, which represent an interesting alternative and open a new reservoir of potential therapeutic phages. There are however issues with the manuscript in the current form, some minor and some more moderate/serious. Detailed comments are provided below, in order as they appeared in the manuscript. In addition, some language editing is recommended.
Line 57. “They then inject their genetic material into the infected bacteria” Bacteria are not infected at that point, please reformulate.
Line 62. “when induced to enter the lytic cycle”. It is the excision of the phage that is induced, please reformulate.
Line 85-88. This should be moved to the result section.
Line 100-105. It is not clear how authors can obtain clones having lost the prophage due to the CI repressor deletion and in the same time clones with modified phage still integrated that can be induced by the addition of mitomycin C. Authors have to comment on that point and give more details.
Line 190. “as expected no amplification was obtained”. This part of the manuscript is problematic since the authors don’t thoroughly verify their constructs (the strain cured from its prophage and Ab105-2phiΔCI). Instead they only do a PCR that do not produce any amplification. Knowing that any unrelated phage would produce similar results with no amplification it is not a proof at all. In addition, off target modifications or induction of other integrated prophages cannot be exclude. For this reason, the strain cured from its prophage and Ab105-2phiΔCI must be sequenced. An alternative would be to design primers out of the CI repressor and prophage integration site in order to have a positive PCR product that can be sequenced to confirm deletion of the repressor deletion and excision of the prophage.
Line 129. “LB supplemented with CaCl2” Is CaCl2 used only for this experiment? Is it an important co-factor for phage adsorption?
Line 143. Not clear how the authors measured the number of phage or antibiotics resistant bacteria. It really must be specified.
Line 176. Please indicate where the treatment was injected.
Line 286. Confirmation has first to be done by sequencing and PCR. See above comment.
Line 293. “including phages from A. baumannii” not clear what it means.
Line 309-310. Authors must define what is their conception of synergism. It is usually admitted that a 2 log more reduction between combined therapy and the compounds alone is synergistic. This is observed in vitro with meropenem and imipenem at ¼ the MIC. However, for the Galleria survival experiment, the fact that combination of phage and antibiotics is more effective than either agent alone does not necessarily mean that the combination has synergistic activity in vivo. It could reflect an additive effect.
Table 1. Not clear what is spot +-. Is it a spot test with phages? ST sequence type?
Table 3. Please specify after how long and at what MOI? Is it in the same condition that in figure A1? The resistance rate is the same for all three antibiotics, also variation is observed in figure 2 regarding bacterial regrowth after 5 hours. Authors should comment on this point. Phage MOI is also missing.
Figure 1. A1. Graphical representation of the Ab105-2phiΔCI phage. Was the mutant really sequenced? A2. It is hard to see the plaques, would need magnification. It would also be interesting to have both Ab105-2phi mutant phage Ab105-2phiΔCI. The quality of the EM image has also to be greatly improved. It is not clear if it is really a bacteriophage or not. Indeed, it can also look like two unrelated particles that are joint by a rode shape one. Since the phage is supposed to be well amplified, authors should have enough opportunities to take a better picture. Moreover, the scale bar is missing, and the picture is too small. Finally, the magnification loop is unrelated and should be removed.
Figure 2. The authors must comment why the phage is more lytic at a MOI of 1 instead of 10, knowing that more phages are present at a MOI of 10. This is unexpected.
Figure 3. Please specify how much phage is injected and what is the inoculum size of. Is the MOI based on inoculum or the number of bacteria infecting larvae at the time of the injection of the phage?
Author Response
The English language was revised and some modifications made in order to improve the manuscript.
Line 57 was changed according as the reviewer suggestion, and “infected” was deleted. Now in line 61. Line 62. The sentence was changed to explain better the lysogenic state of the bacteriophages. Now Lines 64-67. Lines 85-88 were deleted and included into the results section in line 224 and line 241. Line 100-105. As the reviewer suggest, the method of deletion and selection of mutant bacteriophages was explained with more detail, now in lines 100-106. Line 190. As the reviewer says it seems that the deletion is not thoroughly verified. But although we didn´t include this in the manuscript we have done the sequencing and several confirmation PCRs, even those who correspond to a bacterial clon who have lost the phage. All the PCRs were done with primers who amplified 1000 pb for each flanking region of the gene, the CI gene primers and also combinations of them. The PCR amplicons were also sequenced and the deletion confirmed. From the bacterial clon DNA no PCR amplified, so the excision of the phage was confirmed. In this clon, as a control, we also confirmed by PCR the presence of another prophage that was known to be present in the strain (these data are not included in the manuscript). All the information is now included in the Material and method section, lines 103-106, and results lines 204-210. Line 129. The CaCl2 used in all the assays were an infection with the phage was required. We included this in line 140. Also it was previously described in the 2.8 section of the manuscript. Addition of CaCl2 was routinely done in our laboratory as a result of following the protocols described in the literature. Line 143. The method is now explained in detail in lines 147-154. Line 176. Injections were in the left and right prolegs. Now in lines 186 and 189. Line 286. This is corrected in line 306. Line 293. The sentence was corrected. Now in line 314-315. Line 309-310. The definition of synergism is now included in line 324 and also the synergism in galleria was deleted and substituted by the sentence in line 334-335. Table 1 was corrected and explained in the legend. Table 3. The MOI was included in the legend. The conditions are now explained in the material and methods section in lines 147-154. An explanation is now in lines 332-334. Figure 1. The mutant was not sequenced but the sequence corresponds to the wild type phage, for that reason the gene appears with and x. Anyway, the deletion was confirmed by gene sequencing using the flanking regions of CI (now explained in the text). The Figure was modified according to the reviewer instructions and also the legend was modified. Figure 2. The explanation was included in the discussion section in lines 312-313. Figure 3. Both values were included in the legend. MOI was based on bacteria infecting the larvae.Reviewer 2 Report
The manuscript provides an important contribution to the research fields of genetic engineering of phages for therapeutic applications. It will be suitable for publication after the following points have been addressed.
Line 113: 0.22 µm filter instead of 0.22 nm filter?
Table 1: The legend should explain the abbreviation “ST”.
Figure 1B-D: It should be stated in the legend what the graphs represent (mean +/- SD? How many replicates?). Same for figures 2 and 3. The employed statistical tests should also be mentioned in the legends.
Figure 2: In some panels, light blue curves are designated as “C”, in others they are designated as “Control”. Why is an MOI of 10 consistently worse than an MOI of 1, when in Figure 1 B2 an MOI of 10 is more effective?
The entire manuscript should be carefully revised for proper use of the English language and interpunction.
The following articles should be cited and mentioned, perhaps in the Discussion section: PMID: 31465789, PMID: 31123962, PMID: 30687281, PMID: 30532563, PMID: 30237558, PMID: 29760690, PMID: 29755420, PMID: 29687015, PMID: 29375524, PMID: 29258062, PMID: 28974975, PMID: 28807909, PMID: 27431214, PMID: 27208124, PMID: 26925593, PMID: 25411824, PMID: 23071586
Author Response
In order to improve the manuscript the english language was revised and some changes were made.
Line 113. The correct form is now in line 119. Table 1. The legend was modified as suggested. Figure 1 B-D. The legend was modified according to the reviewer instructions. The same was done in figure 2 as in figure 3 this data were previously included. Figure 2. The figure was modified to designate the light blue curves as “Control”. The MOI10 is worse than MOI1 at 6 hours because at this point (as was seen in the infection curves) there are, probably, more phage resistant bacteria, although at first time points (Infection curves) at MOI10 the OD culture was lower than at MOI1. This is now explained in the discussion section in lines 312-313. Some of the citations suggested by the reviewer were included in the manuscript in accordance with the text (PMID: 31465789; PMID: 30237558; PMID: 29760690; PMID: 28807909).Round 2
Reviewer 1 Report
Authors significantly improved the manuscript, which I would recommend for publication in the present form.
Author Response
Thank you for your kind suggestion.